# An In Silico Design of a Vaccine against All Serotypes of the Dengue Virus Based on Virtual Screening of B-Cell and T-Cell Epitopes

**DOI:** 10.3390/biology13090681

**Published:** 2024-08-30

**Authors:** Hikmat Ullah, Shaukat Ullah, Jinze Li, Fan Yang, Lei Tan

**Affiliations:** 1Center for Energy Metabolism and Reproduction, Institute of Biomedicine and Biotechnology, Shenzhen Institute of Advanced Technology, Chinese Academy of Sciences, Shenzhen 518000, China; hikmat@siat.ac.cn (H.U.); ullah@siat.ac.cn (S.U.); 2University of Chinese Academy of Sciences, Beijing 100049, China; 3School of Basic Medicine and Life Sciences, Hainan Medical University, Longhua, Haikou 571199, China; ljz18332352765@163.com; 4Center for Protein Cell-Based Drug, Institute of Biomedicine and Biotechnology, Shenzhen Institute of Advanced Technology, Chinese Academy of Sciences, Shenzhen 518000, China

**Keywords:** Dengue virus, in silico designing, B-cell epitope, T-cell epitopes, pan-serotype vaccine, immune simulation

## Abstract

**Simple Summary:**

Dengue fever is one of the major public health issues in tropical and subtropical regions. To prevent infection by the Dengue Virus, the World Health Organization has approved two dengue vaccines, Dengvaxia and TV003/TV005. But, unfortunately, the vaccines face challenges such as lower effectiveness and antibody-dependent enhancement (ADE). To address these concerns, we have designed a new vaccine, PSDV-2, with optimized B- and T-cell epitopes using an in silico approach. This design prioritizes high antigenicity and immunogenicity while minimizing risks such as allergenicity, toxicity, and ADE. In simulations, PSDV-2 bound strongly to immune receptors TLRs 4 and 2 and triggered a strong immune response. Although further testing is needed to confirm the effectiveness and safety of PSDV-2, our design offers a promising alternative option for dengue fever prevention, potentially contributing to an improvement in global health.

**Abstract:**

Dengue virus poses a significant global health challenge, particularly in tropical and subtropical regions. Despite the urgent demand for vaccines in the control of the disease, the two approved vaccines, Dengvaxia and TV003/TV005, there are current questions regarding their effectiveness due to an increased risk of antibody-dependent enhancement (ADE) and reduced protection. These challenges have underscored the need for further development of improved vaccines for Dengue Virus. This study presents a new design using an in silico approach to generate a more effective dengue vaccine. Initially, our design process began with the collection of Dengue polyprotein sequences from 10 representative countries worldwide. And then conserved fragments of viral proteins were retrieved as the bases for epitope screening. The selection of epitopes was then carried out with criteria such as antigenicity, immunogenicity, and binding affinity with MHC molecules, while the exclusion criteria were according to their allergenicity, toxicity, and potential for antibody-dependent enhancement. We then constructed a core antigen with the selected epitopes and linked the outcomes with distinct adjuvant proteins, resulting in three candidate vaccines: PSDV-1, PSDV-2, and PSDV-3. Among these, PSDV-2 was selected for further validation due to its superior physicochemical and structural properties. Extensive simulations demonstrated that PSDV-2 exhibited strong binding to pattern recognition receptors, high stability, and robust immune induction, confirming its potential as a high-quality vaccine candidate. For its recombinant expression, a plasmid was subsequently designed. Our new vaccine design offers a promising additional option for Dengue virus protection. Further experimental validations will be conducted to confirm its protective efficacy and safety.

## 1. Introduction

Dengue virus (DENV) infection leads to consequent diseases such as dengue fever (DF), dengue hemorrhagic fever (DHF), and dengue shock syndrome (DSS), posing significant global health challenges. Globally, over 50 million people are at risk of DENV infection annually, resulting in approximately 500,000 cases of hospitalization and more than 25,000 deaths. The high prevalence of these diseases imposes substantial socioeconomic and health-related burdens, particularly in tropical and subtropical regions [1,2]. Hence, the World Health Organization (WHO) has recognized the Dengue virus as one of the most dangerous pathogens due to its widespread impact and severity.

Dengue virus, a member of the *Flaviviridae* family, is characterized by a single-stranded, positive-sense RNA genome. The virus is encapsulated in the icosahedral Capsid protein and surrounded by an envelope protein. Its RNA genome, ~11 kb in length, encodes a single polyprotein that is subsequently cleaved into three structural proteins (C, prM, and E) and seven nonstructural proteins (NS1, NS2A, NS2B, NS3, NS4A, NS4B, and NS5). These highly immunogenic viral proteins elicit strong immune responses in the host upon infection [3]. Due to rapid genomic evolution, DENV exhibits significant genetic variations [4], leading to four distinct serotypes (DENV-1, DENV-2, DENV-3, and DENV-4) with approximately 65% sequence similarity [5]. Due to the complexity of antigenic determinants of dengue serotypes, vaccination against a single serotype failed to confer immune protection, i.e., insufficient for complete protection, and even induced certain risks in the host during secondary infection [6]. Thus, a robust balance of immunity across serotypes is crucial for ensuring the safety and effectiveness of dengue vaccines.

In the development of a vaccine to combat DENV, six candidates utilizing various technical platforms have advanced to different preclinical and clinical evaluation stages. These platforms include live attenuated viruses, viral vectors, whole-inactivated viruses, subunit vaccines, and DNA vaccines [7]. Recently, the World Health Organization (WHO) approved two live-attenuated virus vaccines: Dengvaxia, manufactured by Sanofi Pasteur, and TV003/TV005 by Takeda [8]. Both vaccines are tetravalent, designed to provide balanced immunity against all four DENV serotypes. Dengvaxia used a yellow fever vaccine backbone (17D) and incorporated dengue prM and E proteins from all four serotypes. In contrast, TV003/TV005 is attenuated primarily with a 30-nt deletion (rΔ30) in the 3′ untranslated region of the viral genome. Additionally, the chimera component of TV003/TV005 was developed for DENV-2 by replacing the prM and E proteins of DENV-4 rΔ30 with those from the New Guinea C strain. In 2015, Dengvaxia was authorized for clinical application in 20 countries, including the Philippines, Thailand, and Singapore, etc. However, full protection was not achieved, with efficacy varying among serotypes, i.e., 51% for DENV-1, 34% for DENV-2, 75% for DENV-3, and 77% for DENV-4, respectively [9]. Furthermore, concerns arose when severe cases, including deaths, were reported among children under 9 years old who had no previous dengue infection history. TV003/TV005, approved in November 2019 [10], also exhibited lower efficacy, particularly against DENV-2 and 3 [11]. These issues highlight the need for continuing research and development of a highly effective and safe dengue vaccine.

The primary challenge in developing a DENV vaccine is addressing the immune complexities associated with its four serotypes. Due to the genomic similarities among these serotypes, cross-reactive antibodies against the envelope (E) protein can become prominent during secondary infections [12]. Unfortunately, these antibodies, while potent in binding to the E antigen, facilitate viral entry rather than provide protective immunity across serotypes. This phenomenon, known as antibody-dependent enhancement (ADE), often results in severe conditions such as dengue hemorrhagic fever or dengue shock syndrome [13,14]. To develop a safe and effective dengue vaccine, it is crucial to identify and eliminate epitopes that may contribute to ADE. Additionally, achieving a balanced immune response by inducing comparable antibody levels across all four serotypes is essential for providing comprehensive protection.

Advances in immunoinformatics have made computational vaccine design feasible and efficient. In silico prediction can identify B-cell, CTL, and HTL epitopes while avoiding allergens and other side effects, enhancing vaccination efficiency and safety. Recent proposals for in silico Dengue virus (DENV) vaccine construction have emerged [15,16,17,18,19]. However, current designs face challenges, including covering the diverse dengue genome with a limited number of epitopes and managing molecular size. Most designs still use a tetravalent composition, complicating manufacturing and validation, and often overlook ADE-associated epitopes.

Therefore, in our study, we addressed these issues with the following strategies: **Conserved Fragments:** Identified conserved regions in the viral polyprotein to form the basis of epitope selection;**ADE Risk Exclusion:** Excluded potential ADE-associated epitopes based on a comprehensive literature review;**Serotype-Specific Epitopes:** Combined serotype-specific B-cell epitopes from the E protein with pan-serotype T-cell epitopes from other proteins, including NS1, NS3, NS5, and Capsid;**Molecule Integration:** Integrated all selected epitopes and adjuvant proteins into a single, reasonably sized molecule rather than a complex tetravalent formulation.

Prospectively, this strategy aims to offer a new dengue vaccine design with improved effectiveness, safety, and feasibility.

## 2. Materials and Methods

In the current study, we aimed to design a novel DENV vaccine that is efficacious against all four serotypes. The complete design process is illustrated in Figure 1. The web addresses for the databases and platforms used in this study are listed in Appendix A. All supplementary tables are included in Appendix A, and supplementary figures are provided in Appendix A.

### 2.1. Retrieval and Analysis of Full-Length Dengue Polyprotein Sequences

To initiate the vaccine design process, full-length polyprotein sequences of DENV serotypes 1 to 4 were obtained from 10 representative countries and four reference strains (Appendix A) via the NCBI (National Center for Biotechnology Information) and VIPR (Virus Pathogen Database and Analysis Resource) databases. Within each serotype, conserved fragments were identified by performing multiple sequence alignments (MSA). The primary alignment was performed with MUSCLE (https://www.ebi.ac.uk/jdispatcher/msa/muscle?stype=protein, accessed on 11 August 2024) [20], with subsequent verification through CLUSTALW version 2.0 [21] and MAFFT (https://mafft.cbrc.jp/alignment/server/, accessed on 11 August 2024) [22]. Conserved fragments were then extracted using BioEdit 7.2 [23], as detailed in Appendix A. Phylogenetic analyses were conducted to elucidate the evolutionary relationships among DNV strains. Moreover, evolutionary distances were calculated with the Molecular Evolutionary Genetics Analysis (MEGA) 4 software package, elucidating the evolutionary distance among DENV strains [24]. A phylogenetic tree was constructed using the neighbor-joining method [24] and visualized with the Interactive Tree Of Life (iTOL) [25].

### 2.2. Step-by-Step Selection of Epitopes and the Estimation of the Coverage in the Population

Epitopes for B Cells, CTLs, and HTLs were meticulously identified from the conserved sequences of DENV-1 through DENV-4 polyproteins employing a rigorous immune-informatics approach.

#### 2.2.1. Identification of B-Cell Epitopes

**Prediction:** Continuous B-cell epitopes were predicted in conserved sequences of DENV-1~4 polyproteins using the IEDB database with the Bepipred2.0 method [26]. It utilizes a hidden Markov model to predict linear B-cell epitopes based on the peptide’s propensity to bind to antibodies. A threshold of 0.5 was set to determine the likelihood of an epitope being immunogenic;**Selection criteria:** Epitopes were first evaluated for their antigenicity (their ability to elicit an immune response) with the online tool VaxiJen [27], and then intra-serotyped conservancy with the IEDB database to ensure they were preserved within the same serotype of DENV. After that, allergenicity and toxicity were evaluated using Allertop v.2.0 [28] and Toxinpred [29] respectively;**Exclusion of ADE-related epitopes:** To mitigate the risk of ADE, relevant literature was collected and utilized to extract the potentially problematic epitopes, detailed in Appendix A. ADE-related epitopes were then excluded accordingly from the list of predicted B-cell epitopes, mainly based on their sequence and location in the envelop protein;**Conservancy analysis:** The intra-serotype and cross-serotype conservancy of the epitopes were further analyzed to ensure they were effective across different strains within the same serotypes but not among distinct serotypes. The conservancy analysis was perfumed using the IEDB-based conservancy analysis server [30].

#### 2.2.2. Identification of CTL and HTL Epitope

**Prediction of epitopes in given length:** Epitopes were predicted using NetMHC 4.0 [31] for CTLs (MHC I) and NetMHC MHC class II 2.3v for HTLs (MHC II) [32]. These tools predict binding affinities between peptides and MHC molecules, which are critical for T-cell activation. For CTL epitopes, the length was defined as 9 amino acids, while HTL epitopes were considered 15 amino acids. These lengths corresponded to the peptide fragments that can be effectively represented by MHC molecules;**Binding affinity-based selection of epitopes:** Primary selection was conducted based on the binding affinity to MHC I/II receptors, with strong binders (SB) being those in the top <0.5% of the predicted binding scores for both CTLs and HTLs, while weak binders (WB) in the top <2% for either type. This ensured that the selected epitopes had a high likelihood of inducing a strong immune response;**Further criteria of epitope evaluation:** Epitopes with strong binding affinity were further evaluated for their antigenicity, immunogenicity, and intra- and cross-serotype conservancy while ensuring they did not possess characteristics of toxicity or allergenicity.

#### 2.2.3. Population Coverage Analysis

To ensure that the vaccine design was effective across diverse populations, the selected MHC I/II epitopes were analyzed for global population coverage using the IEDB population coverage tool [18]. All global regions were incorporated into the analysis with both classes of MHC. 

### 2.3. Formulation and Evaluation of Vaccine Candidates

To formulate candidate vaccine molecules, selected epitopes were grouped corresponding to B cells, CTLs, and HTLs, respectively, and then assembled into a vaccine construct in conjugation with adjuvant protein domains through the utilization of linkers [33]. 

Two adjuvant proteins were employed to boost the immune response. One was Human beta-defensin 3, known for its antimicrobial and immunomodulatory properties, and the other was Heparin-binding Hemagglutinin (HBHA), derived from the *Mycobacterium* sp. [34,35]. For further enhancement of immunogenicity of the vaccine candidate, the pan-HLA DR binding epitopes sequence (PADRE, 13aa) was incorporated additionally to stimulate CD4^+^helper T cells [33,36].

The following linker sequences were used in the vaccine assembly:**B-cell epitopes:** A flexible linker sequence, KK, was used to connect B-cell epitopes [37];**CTL epitopes:** A different flexible linker, AAY, was employed for the connection between CTL epitopes;**HTL epitopes:** A glycine–proline-rich linker, GPGPG, was used to link HTL epitopes;**Inter-group linkers:** A flexible GGGS linker was utilized to join distinct groups of epitopes, ensuring a cohesive antigen structure [33];**Adjuvant integration:** Adjuvant proteins were conjugated to the N- or C-terminus of the epitope groups via a rigid EAAAK linker.

With the formulated candidate vaccines, a panel of immunological properties was assessed. To ensure the ability to induce immunity, VaxiJen and ANTIGEN pro servers were adopted to score their antigenicity. A threshold of 0.4 was applied [27]. Moreover, allergenicity was evaluated using Allertop v.2.0l [38] to make sure the vaccine did not elicit an allergic reaction. Whereas toxicity was analyzed with Toxinpred [29] to identify and mitigate any potential toxic effects of the vaccine. 

The solubility of vaccine molecules was analyzed with Protein-Sol server [39] and SOLpro [40] to assess the solubility of the vaccine molecules, which is critical to their manufacture and application. Further characterization was conducted using the Expasy Protparam server for theoretical calculation of pI (isoelectric point), molecular weight, instability index, and grand average of hydropathicity GRAVY [41].

### 2.4. Predictions and Validation of Molecular Structure for Vaccine Candidates

The secondary structure of the vaccine protein was predicted using the PSIPRED online server [42]. This server employs a neural network-based approach to analyze the amino acid sequence and predict the secondary structural elements, including α-helices, β-strands, and coils.

The tertiary structure of the vaccine protein was predicted by submitting the protein sequence to the PHYRE-2 protein folding recognition server [43]. PHYRE-2 utilizes advanced algorithms for protein structure prediction, such as homology modeling and fold recognition, to generate three-dimensional models of the proteins.

The best predicted 3D models were superimposed and analyzed for conformational changes using the UCSF Chimera (https://www.cgl.ucsf.edu/chimera/) [15]. This software allows the visualization and comparison of the structural models, enabling the assessment of conformational alterations and model accuracy.

Additionally, to evaluate the stereo-chemical quality of the predicted protein structure, Ramachandran plots were generated using the PROCHECK online server [44]. This analysis assesses the dihedral angles of amino acid residues to verify that they are within favorable regions of the Ramachandran plot.

The ProSA-web [45] was employed to further validate the 3D structure of the modeled vaccine. ProSA-web provides a comprehensive assessment of the structural quality by evaluating the overall energy and potential errors in the predicted model. 

By integrating these structural prediction and validation methods, the vaccine protein’s secondary and tertiary structures were accurately modeled and assessed, ensuring reliability in further experimental analyses and applications.

### 2.5. Molecular Docking and Dynamics of the Vaccine-Immune Receptor Complexes

The binding of vaccine protein with pattern recognition receptors, such as Toll-like receptors (TLRs) or MHC molecules, was evaluated using AMBER18 for the molecular docking and Hawk Dock for the binding energy calculations [18]. These tools facilitate the prediction of binding affinities and the characterization of the interaction interface of antigen-immune receptor complexes in detail. The resulting models were visualized and assessed using the UCSF Chimera tool [46]. Prevalent human leukocyte antigen (HLA) alleles and TLR structures were retrieved from the Protein Databank (PDB) (the PDB IDs are listed in Appendix A) [18,33]. Analysis of non-covalent interactions, including hydrogen bonds, salt bridges, and interface areas, was conducted using PDBSum [47].

Molecular dynamic (MD) simulations were performed for the top-ranked vaccine receptor complexes involving TLR2 and TLR4 using the AMBER18 simulation platform [48]. The system was configured with 12-angstrom padding around the protein and water box. The TLR receptors utilized the ff14SB force field [49,50], while vaccine molecules were parameterized using the GAFF force field [51]. The vaccine molecules were placed in a TIP3P water box, and counter ions were added to neutralize the system. The system underwent minimization using the steepest descent or conjugate gradient methods, with transition steps every 1000 cycles up to a maximum of 5000 cycles to reduce structural collisions. Following minimization, the system was heated over 250 picoseconds (ps) to reach thermal equilibrium at 300 K. A production run of 100 nanoseconds (ns) was conducted to explore the dynamic behavior of the vaccine–receptor complexes. The SHAKE algorithm [52] was applied to maintain the rigidity of covalently bound hydrogen atoms, and periodic boundary conditions were used within the canonical ensemble. The temperature was regulated at 300 K using a Langevin thermostat, with a 10 Å cutoff for non-bounded interactions and Ewald simulations for long-range interactions. The CPPTRAJ module [53] was used to analyze statistical parameters such as root mean square deviation (RMSD), root mean square fluctuations (RMSF), radius of gyration (Rg), and hydrogen bonds of the vaccine–receptor complex. The results were visualized with Pymol [54] to interpret the structural stability and dynamic properties of the vaccine–receptor complexes. These methodologies provided a comprehensive evaluation of the vaccine protein’s binding interactions with immune receptors and its dynamic behavior, aiding in the design and optimization of effective vaccine candidates.

### 2.6. Immune Simulation of Selected Candidate Vaccine

The immune simulations were performed using the C-ImmSim server to model the effects of vaccine administration and subsequent immune responses [55]. The simulation was designed to reflect a vaccination schedule comprising three doses administered at 2-week intervals. The time points for these doses were set at simulation steps 1, 84, and 168, where each time step corresponded to an 8-h interval in real-time [56]. All other simulation parameters were maintained at their default settings to ensure standardization and consistency in the immune response modeling.

This approach provided insights into the immune system’s reaction to the vaccine over the course of the dosing schedule, helping to predict the vaccine’s efficacy and immune activation profile.

### 2.7. Codon Optimization and In Silico Cloning of Selected Candidate Vaccine

The peptide sequence of the candidate vaccine was submitted to the JCat server for codon optimization, with *E. coli* K12 selected as the preferred host organism [57]. The optimization process adhered to several key guidelines to enhance expression efficiency in *E. coli*: (1) elimination of restriction enzyme cleavage sites, (2) avoidance of prokaryotic ribosome binding sites, and (3) exclusion of transcription rho-independent termination sites. The optimized sequence was evaluated based on a codon adaptation index (CAI) greater than 0.8 and a GC content of 30 to 70% [57] to ensure effective expression and stability. Subsequently, the modified nucleotide sequence was integrated into the pET28a (+) expression vector through an in silico cloning process. This procedure was facilitated using SnapGene version 4.2 [58], which allowed for precise vector construction and sequence verification.

## 3. Results

### 3.1. Conserved Fragments Were Extracted from DENV Polyprotein Sequences Collected Worldwide

Viral sequences were selected based on the epidemiological relevance, geographical distribution, and data availability from 10 representative countries: Bangladesh, Brazil, China, India, Pakistan, Philippines, Singapore, Thailand, the U.S.A, and Vietnam. A total of 2929 full-length polyprotein sequences of DENV from these countries, spanning the years 1960 to 2023, were downloaded. Specifically, the number of sequences per serotype was as follows: DENV-1 (1355), DENV-2 (1031), DENV-3 (400), and DENV-4 (143). The detailed numbers of sequences for each serotype collected from each selected country are provided in Appendix A, while the accession numbers of each polyprotein sequence are listed in Appendix A.

The collected sequences were aligned to identify conserved regions and construct phylogenetic trees for each serotype. Phylogenetic analysis revealed an overall mean distance of 0.02, indicating close genetic relationships within each serotype (Appendix A). The phylogenetic trees showed that sequences were predominantly clustered by serotype, with sequences from the same region or country exhibiting closer genetic similarities, suggesting common ancestral origins. Based on the alignments, conserved fragments were identified for each serotype, and the ones no less than 15aa were used in subsequent epitope screening (Appendix A). In brief, 63 fragments were harvested for DENV-1, 71 for DENV-2, 72 for DENV-3, and 69 for DENV-4. 

### 3.2. Effective and Non-Risky Epitopes Were Selected Stepwise from Conserved Fragments of DENV Polyprotein for B Cells, CTLs, and HTLs

B-cell epitopes were initially identified using a threshold of 75% sensitivity and 50% specificity, yielding a total of 113 epitopes for DENV-1 and 92, 92, and 85 epitopes for DENV-2, 3, 4, respectively (Appendix A). The initial list of epitopes was refined based on an antigenicity score ≥ 0.4, non-allergenic, non-toxic, with an intra-serotype conservancy and without a cross-serotype conservancy. This process significantly reduced the number of candidate epitopes to four for DENV-1 and three each for DENV-2, 3, and 4 (Appendix A). Further refinement involved the exclusion of epitopes associated with ADE-associated epitopes, as identified in previous studies [54,55]. The literature supporting this exclusion is listed in Appendix A. Attention was paid to the regions within domains I and II of the envelope protein, which have been reported as being highly correlated with ADE effects [59]. Therefore, the candidate B-cell epitopes located in these two regions were largely removed from the list. Finally, following all filtering steps, one epitope per serotype was selected for inclusion in the final vaccine design (Table 1). The selected B-cell epitopes were mapped to the surface of the envelope protein, with epitopes for DENV-1 through DENV-3 located in domain III and the epitope for DENV-4 located in domain II (Appendix A), which was to ensure their accessibility for immune recognition.

For CTLs or HTLs epitopes, initial predictions generated extensive lists with over 2000 potential epitopes per serotype. These epitopes were first filtered based on binding affinity, retaining those with IC_50_ value ≤ 50 nM. This process shortened the list to approximately 30~60 epitopes per serotype for each class (Appendix A for MHC-I and Appendix A for MHC-II; Appendix A). Subsequently, the epitopes were ranked based on their binding affinity, antigenicity, and immunogenicity for MHC I, and binding affinity and antigenicity for MHC II. Epitopes were further excluded based on allergenicity and toxicity. As the resultant final selection, 11 epitopes were selected for MHC I (CTLs) from nonstructural proteins NS1, 3, and 5 (Table 2), while 7 epitopes were chosen from the structural protein C for MHC II (HTLs) (Table 3, and Appendix A). Notably, while cross-serotype conservancy was minimal among MHC I epitopes, it was prevalent among MHC II epitopes (Table 2 and Table 3). The population coverage with the selected HLA alleles was estimated to be 62.26% (Appendix A).

### 3.3. Three Candidate Vaccines Composed of Selected Epitopes Were Predicted as Stable, Water Soluble, and Antigenic

The core antigen for each vaccine candidate was designed by assembling the selected epitopes for B cells (4), CTLs (11), and HTLs (7) into a unified peptide sequence. These epitopes were linked with peptide linkers and flanked by two copies of the PADRE peptide to enhance immunogenicity. The resulting core antigen was then conjugated with different adjuvant proteins to create three distinct vaccine candidates, PSDV-1 through PSDV-3, as outlined below:PSDV-1: Assembling the core antigen with Heparin-binding Hemagglutinin (HBHA) at the N-terminus and beta-defensin at the C-terminus;PSDV-2: Assembling the core antigen with HBHA at the N-terminus only;PSDV-3: Assembling the core antigen with beta-defensin at the N-terminus only.

Figure 2 illustrates the schematic representation of these vaccine constructs, highlighting the arrangement of epitopes and adjuvant proteins. Peptide sequences of PSDV-1~3 are presented in Appendix A.

For the size of PSDV-1~3, peptide length was calculated as 577aa, 508aa, and 394aa (Line 1 in Table 4), while molecular weight was 63.4 kDa, 55.3 kDa, and 42.7 kDa (Line 2 in Table 4), respectively.

Their stability was estimated using three independent parameters: the Instability Index, Aliphatic Index, and half-life integrally. All candidates demonstrated an Instability Index value below the threshold of 40, listed as 25.25, 24.28, and 20.52 for PSDV-1~3, respectively (Line 6 in Table 4). Their Aliphatic Index was calculated as 86.55, 88.33, and 85.056, respectively (Line 7 in Table 4). The intracellular half-life was estimated to be more than 30 h in mammalian cells for all three molecules (Line 5 in Table 4). All these analyses exhibited good stability for PSDV-1~3 in vitro and in vivo, suggesting they are promising candidates for further evaluation and development.

Good solubility in water was suggested for PSDV-1~3 with the Solubility Index of about 0.79, 0.76, and 0.56, respectively (Line 12 in Table 4), which is favorable for their formulation and delivery. In addition, the GRAVY (Grand average of hydropathicity) predicted for PSDV-1 was −0.027, PSDV-2 scored 0.018, and PSDV-3 scored 0.151. As suggested by the GRAVY value, PSDV-1 was hydrophilic, favoring interaction with aqueous environments; PSDV-2 was nearly neutral, indicating balanced hydrophilic and hydrophobic properties, and PSDV-3 was hydrophobic, which may influence its interaction with different environments and potential cellular uptake (Lines 9-11 in Table 4). Based on theoretical pI, PSDV-1 and -3 were weakly alkaline proteins, and PSDV-2 was weakly acidic (Line 3 in Table 4). 

Moreover, all three vaccine candidates (PSDV-1, PSDV-2, and PSDV-3) were evaluated as antigenic, non-allergenic, and non-toxic (Line 6 in Table 4). These findings suggested that the candidates were likely immunogenic without eliciting adverse allergic or toxic responses, making them suitable for further development and testing.

### 3.4. PSDV-2 Was Highlighted for Further Evaluation Based on Structural Advantages and Showed a Tight Interaction with TLRs and HLAs

The structural analysis of the PSDV-1, PSDV-2, and PSDV-3 vaccine candidates revealed key differences in their secondary and tertiary structures. The secondary structure composition showed that PSDV-1 comprised 31.71% coils, 56.67% helices, and 11.61% β-strands; PSDV-2 had 33.46% coils, 55.70% helices, and 10.82% β-strands, while PSDV-3 contained 29.08% coils, 37.81% helices, and 23.10% β-strands (Appendix A and Figure 3A,B). Tertiary structure modeling using Phyre2, with templates 5d4w, 6tpi, and 1kj6 for PSDV-1, PSDV-2, and PSDV-3, respectively, provided confidence levels of 87.9%, 97.4%, and 99.9%, with coverage of 47%, 59%, and 11% (Appendix A and Figure 3C–F). Ramachandran plots indicated that PSDV-2 had the highest overall quality with 97.6% of residues in favored (92.0%) and allowed (5.6%) regions (Appendix A and Figure 3G), compared to PSDV-1 and PSDV-3, which had 98.3% (85.2% in favored region plus 13.1% in allowed region) and 89.5% (76.3% in favored region plus 13.2% in allowed region) of residues in these regions, respectively, indicating that PSDV-2 had the highest overall quality score and favored regions among the three candidates (Appendix A and Figure 3H). ProSA analysis further supported PSDV-2’s superior quality with a Z-score of −5.19 (Figure 3I), the most favorable among the candidates, compared to PSDV-1 and -3, which presented as −3.73 and −4.75, respectively (Appendix A and Figure 3J). These results suggested that PSDV-2 had the most optimal structural properties, making it the best candidate for further evaluation.

Molecular docking of the PSDV-2 vaccine candidate was performed using Hawk Dock to evaluate its interaction with four HLA alleles (HLA-DR B101:01, HLA-DRB104:01, HLA-A0201, and HLA-DRB10701) in Appendix A and two TLRs (TLR2: Figure 4A,B, TLR4: Figure 4E,F). The conformation with the lowest binding energy was selected for further analysis (Appendix A and Figure 4). The binding energy for these complexes ranged from 5000 to 7000 kcal/mol, with the lowest values for TLR2 and 4 being −7330.87 and −5386.23 kcal/mol, respectively. Moreover, Figure 4, Appendix A, and Table 5 show that the interaction between PSDV-2 and TLR2 mediated four hydrogen bonds and three salt bridges (Figure 4C,D), with the binding interface composed of 20 residues in PSDV-2 (1023Å^2^) and 21 residues in TLR2 (1066Å^2^). Conversely, the binding of PSDV-2 with TLR4 involved five hydrogen bonds and three salt bridges being formed, with 13 residues in the vaccine PSDV-2 (893Å^2^) and 14 residues in the TLR4 receptor (Figure 4G,H), (877Å^2^) involved in the shaping of participating in the interface formation.

The conformational stability of PSDV-2–TLR complexes was then confirmed in molecular dynamic simulations with 100 ns intervals (Figure 5). Key metrics are illustrated as follows:**Root mean square deviation (RMSD):** The RMSD values for either PSDV-2–TLR4 or PSDV-2–TLR2 complexes remained consistently within the allowable range of 4 Å, and reached equilibrium within approximately 40 ns, indicating stable conformational behavior;**Root mean square fluctuation (RMSF):** As a reflection of the flexibility of residues, RMSF values for either complex were stabilized within the permissible 4 Å range, suggesting minimal fluctuations and consistent structural integrity;**Radius of gyration (Rg):** As a measure of the compactness of the protein structure, the Rg value was approximately 37 Å for the PSDV-2–TLR2 complex and 34 Å for the PSDV-2–TLR4 complex, indicating well-maintained structural compactness;**Hydrogen bonds:** The number of hydrogen bonds formed within the TLR receptors remained between 20 and 25 throughout the simulation, highlighting a strong and consistent interaction.

These results collectively demonstrate the robust and stable interaction between PSDV-2 and TLR2/4 receptors, confirming that PSDV-2 maintained effective recognition and binding by the host immune system, which is the key step to initiating host immune responses.

### 3.5. Robust Responses Were Induced for both Innate and Adaptive Immunity upon the Simulation of the PSDV-2 Candidate Vaccine

The effectiveness of the PSDV-2 candidate vaccine was evaluated using the C-ImmSim platform with three doses and 2-week intervals. 

For innate immunity, the status of DCs, macrophages, and NK cells are presented in Figure 6B–D, and the activation of macrophages was clearly observed after immunization. Moreover, the production of key cytokines was robustly induced with PSDV-2 (Figure 6A). The level of IFN-γ and IL-2 was particularly high, with the peak value exceeding 400,000 ng/mL, highlighting a robust immune response (Figure 6A). These data underscore PSDV-2’s effectiveness in stimulating both innate immune cells and significant cytokine production.

Activation of the adaptive immune system was also triggered by the virtual administration of PSDV-2 (Figure 7). Following the administration of three doses at two-week intervals, the vaccine elicited a strong and sustained activation of B cells (Figure 7B) and memory helper T cells (Figure 7F). B-cell activation was notably robust, with significant increases in both IgM isotypic B cells and memory B cells (Figure 7B), which remained elevated for nearly one year post-vaccination. The response was characterized by a peak in antibody titers, with IgM and IgG levels surpassing 140,000 (Figure 7G), indicating a potent humoral response range similar to other reported in silico vaccine designs [60]. The major antibodies produced were predominantly IgM, followed by IgG1, with IgG2 present in smaller quantities (Figure 7G). The generation of these antibodies, alongside total B-cell activation, reached their peaks no later than 50 days after the first dose (Figure 7A,G). This was slightly earlier than the peak observed for cytotoxic T lymphocytes (CTLs) (Figure 7C). Additionally, a significant boost in memory helper T cells was observed, maintaining elevated levels for over 350 days (Figure 7F). These data collectively demonstrate the vaccine’s effectiveness in stimulating and sustaining both the humoral and cellular arms of the adaptive immune system. 

In one word, inoculation with the PSDV-2 candidate vaccine will immunize systematically, establishing powerful protection against DENVs and fostering long-term immune memory in T and B cells.

### 3.6. Codon Optimization and In Silico Cloning of the Designed Vaccines

For the efficient production of the PSDV-2 vaccine, a recombinant expression system using *E. coli* was employed. A DNA sequence of 1524 base pairs (bps) was synthesized based on the designed peptide sequence. This sequence underwent codon optimization using the JCAT (Java Codon Adaptation Tool), achieving a codon adaptation index of 0.97 to ensure compatibility with the K12 strain’s cellular machinery. The GC content of the optimized sequence was approximately 54.65%. To facilitate cloning, restriction sites SgrAI and PstI were added to the 5’ and 3’ ends of the sequence. This optimized sequence was then inserted into the pET28a (+) vector in silico, resulting in the creation of an expression plasmid with a total length of 6893 bps (Figure 8).

## 4. Discussion

The quest to develop an effective vaccine against the Dengue virus (DENV) has been ongoing for over 80 years; yet a definitive treatment remains elusive. The journey toward a successful vaccine has been arduous, with several approaches, including inactivated vaccines [61], virus-like particle-based vaccines [62], and subunit vaccines [63], Despite these efforts, clinical trials have revealed significant limitations. For instance, Dengvaxia [57] and TV-003/TV005 [7], both advanced vaccine candidates, demonstrated inadequate induction of robust humoral immunity, particularly against DENV-2. In the case of Dengvaxia [64], it resulted in severe adverse outcomes, including fatalities in some cases. These challenges highlight intrinsic risks associated with live attenuated vaccines, such as the potential for viral reversion to a wild-type phenotype or unexpected recombination events in vivo. Live attenuated vaccines, like Dengvaxia, retain a full complement of viral proteins, including those associated with antibody-dependent enhancement (ADE) effects or other risks, which complicates their safety profile. Consequently, there is an urgent need for novel vaccine strategies that can achieve efficient, long-lasting, and robust adaptive immunity while minimizing potential risks associated with ADE and other vaccine-related complications.

For this purpose, a computer-assisted multi-epitope vaccine design has emerged as a promising alternative when compared to conventional strategies based on wet experiments. Since RinoRappuoli pioneered the concept of reverse vaccinological immunoinformatic design in 2000 [65], and applied it to the design of a vaccine against Serogroup B meningococcus [66], this approach has been employed for a variety of pathogens. Its applications include vaccines for the Chikungunya virus [67], Saint Louis encephalitis virus [68], SARS-CoV-2 virus [69], Theileriaparasite [70], and also the Dengue virus [71]. Since 2009, multiple proposals for Dengue virus (DENV) vaccines have been published utilizing in silico methods. These designs have focused on selecting effective epitopes while predicting and excluding those with the potential for allergic or toxic reactions [16,18,33,72]. For instance, an early in silico vaccine design based on the E protein of DENV-2 and the backbone of DENV-3 aimed at targeting all four DENV serotypes, though it was criticized for its impracticality and lack of subsequent immune simulation [72].

More recent studies from 2016 to 2023 have continued this trend, but our approach marks a significant departure from previous efforts. Unlike earlier designs that utilized representative sequences of full-length polyproteins or individual viral proteins [15,17,72,73,74,75], our strategy begins by extracting conserved fragments from the full-length sequences of DENV polyproteins sourced from ten geographically diverse countries. This method addresses the extensive genetic diversity among DENV strains, which encompasses four serotypes, numerous genotypes, and thousands of clinical isolates. By focusing on conserved fragments, we aimed to develop a vaccine that provides broad-spectrum protection with a refined selection of well-characterized epitopes, thus enhancing the likelihood of inducing effective and durable immunity across different DENV strains.

The second key aspect of our strategy involved comprehensive epitope screening across all 10 viral proteins of DENV. Previous in silico designs [18,33] have similarly utilized the full complement of DENV proteins, demonstrating the potential for broad-spectrum immunity based on computational simulations. Experimental studies have further elucidated the functional roles of different viral proteins in immune response. For example, the E III domain has been identified as containing crucial determinants for neutralizing antibodies [76]. Additionally, CD8+ T cell (cytotoxic T lymphocyte, CTL) epitopes are predominantly located in the nonstructural proteins NS1, NS3, and NS5. In contrast, CD4+ T cell (helper T lymphocyte, HTL) epitopes are largely found within the structural protein C, as well as proteins E and NS1, which also play a significant role in B-cell activation and antibody production [77]. In our design, the selection of epitopes was carefully aligned with these findings to maximize the efficacy of immune induction. By focusing on these key protein regions known to be involved in critical immune responses, we aimed to enhance the overall effectiveness of the vaccine. This approach ensured that the selected epitopes were optimally positioned to stimulate both humoral and cellular immune responses, providing a robust and comprehensive immune defense against the diverse strains of DENV.

The final set of epitopes identified included 4 specific to B cells, 11 for CD8+ CTLs, and 7 for CD4+ HTLs. Each of these epitopes was rigorously evaluated through a stepwise screening process to ensure they were antigenic, immunogenic, non-allergenic, and non-toxic. This process involved additional scrutiny to eliminate potentially risky components. Specifically, epitopes associated with ADE were meticulously reviewed and removed if they exhibited cross-serotype conservancy or were situated in regions suspected of contributing to ADE. This precautionary measure aimed to minimize ADE potential based on current knowledge and to avoid any adverse effects on immune balance. For T-cell epitopes, we further refined the selection by evaluating their binding strength to MHC class I and II receptors. Prioritization was given to epitopes demonstrating high cross-serotype conservancy, which is expected to enhance protection across different DENV serotypes. This strategy culminated in the design of a tetravalent core antigen that integrated epitopes from all four serotypes into a single molecule. This consolidated approach not only maintained a manageable size but also aimed to provide comprehensive coverage and induce both humoral and cellular immune responses against all DENV serotypes.

Subsequently, three candidate vaccine molecules—PSDV-1, PSDV-2, and PSDV-3—were constructed by linking the core antigen with distinct adjuvant protein domains: PADRE, HBHA, and beta-defensin, respectively. Among these, PSDV-2, which combined PADRE with HBHA at the N-terminus, was selected for further evaluation due to its advantageous structural properties. Molecular docking simulations demonstrated that PSDV-2 formed stable complexes with pattern recognition receptors, including TLR2, TLR4, and HLAs. These interactions suggest that PSDV-2 will be effectively recognized as a foreign antigen by the host immune system, a crucial step for eliciting a robust immune response.

Consequently, in silico simulations indicated that the designed vaccine has the potential to provide comprehensive immune protection against DENV. In terms of innate immunity, the vaccine is predicted to stimulate the secretion of cytokines and enhance the activity of NK cells, DCs, and macrophages. These effects are crucial for initiating and regulating the subsequent adaptive immune response. Following three doses of immunization, the simulations projected a significant generation of antibodies, with antibody titers reaching 1:15,000, comparable to levels observed in previous vaccine designs. Additionally, the vaccine is expected to activate B cells, CD4+ HTLs, and CD8+ CTLs, while establishing long-lasting immune memories in both B cells and HTLs.

The estimated global population coverage of this vaccine was approximately 62.26%, which is competitive compared to other vaccine candidates.

In summary, the current study presented a promising candidate vaccine designed to offer protection against all four DENV serotypes. The vaccine molecule was optimized for size, stability, and physicochemical properties, ensuring robust immune protection. Notably, our design strategy included two distinct innovations compared to previous approaches: (1) Epitope screening was conducted on conserved fragments derived from a global collection of DENV protein sequences, and (2) Risky elements, particularly ADE-associated B-cell epitopes, were systematically excluded. These considerations collectively enhance both the efficacy and safety profile of the candidate vaccine, positioning it as a potentially effective solution for combating dengue fever worldwide.

## 5. Conclusions

Our design of a multi-epitope vaccine against the Dengue virus (DENV) utilized an efficient and methodical immunoinformatics approach. The process involved a detailed screening and elimination of potentially harmful epitopes to develop the PSDV-2 candidate vaccine. This vaccine incorporated B-cell epitopes from E protein domains II and III, CTL epitopes from nonstructural proteins NS1, NS3, and NS5, and HTL epitopes from the C protein. It was further enhanced with HBHA and PADRE adjuvants to boost immunogenicity. Comprehensive simulations demonstrated that PSDV-2 possessed favorable molecular properties, effectively interacts with Toll-like receptors (TLRs) and Human Leukocyte Antigens (HLAs), and is projected to provide robust immune protection for over 60% of the global population. These findings underscore the potential of PSDV-2 as a viable and practical vaccine candidate for DENV. Nonetheless, validating its effectiveness and safety through further in vitro and in vivo experiments remains essential.

## Figures and Tables

**Figure 1 biology-13-00681-f001:**
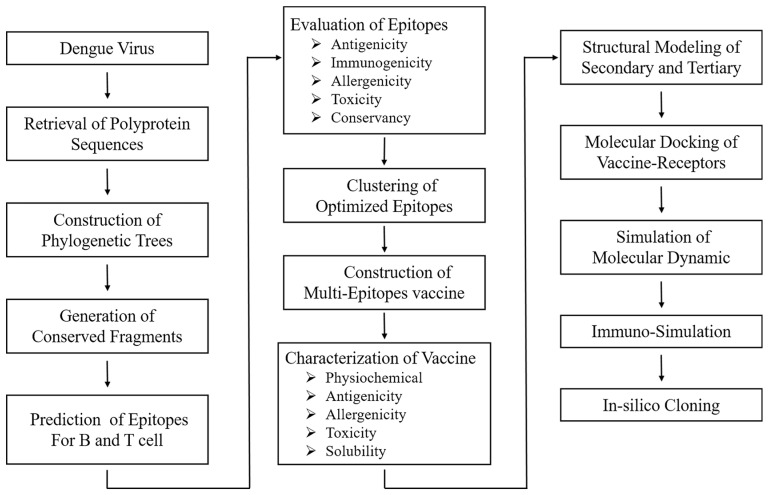
Flow chart of the Dengue vaccine design process using reverse vaccinology. The key stages are highlighted, from data collection and epitope prediction to vaccine candidate construction and in silico validation.

**Figure 2 biology-13-00681-f002:**
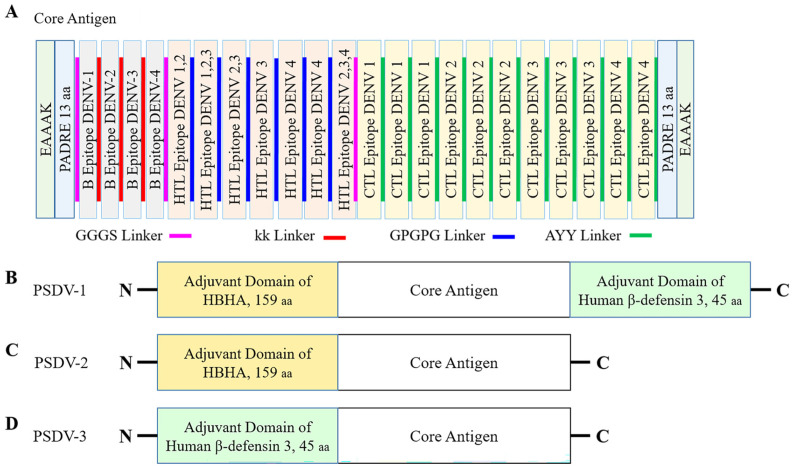
**Organization of multi-epitope tetravalent vaccine peptides.** (**A**) Graphic presentation of the vaccine design strategy: core antigen; Overview of the strategy used for constructing the multi-epitope tetravalent vaccine peptides. The design incorporates selected epitopes for B cells, CTLs, and HTLs, linked with peptide linkers and flanked by PADRE peptides. (**B**–**D**) Vaccine candidates PSDV-1, PSDV-2, and PSDV-3. (**B**) PSDV-1: Core antigen conjugated with Heparin-binding Hemagglutinin (HBHA) at the N-terminus and beta-defensin at the C-terminus. (**C**) PSDV-2: Core antigen with HBHA at the N-terminus only. (**D**) PSDV-3: Core antigen with beta-defensin at the N-terminus only. Each panel illustrates the specific arrangement of epitopes and adjuvant proteins in the respective vaccine candidate.

**Figure 3 biology-13-00681-f003:**
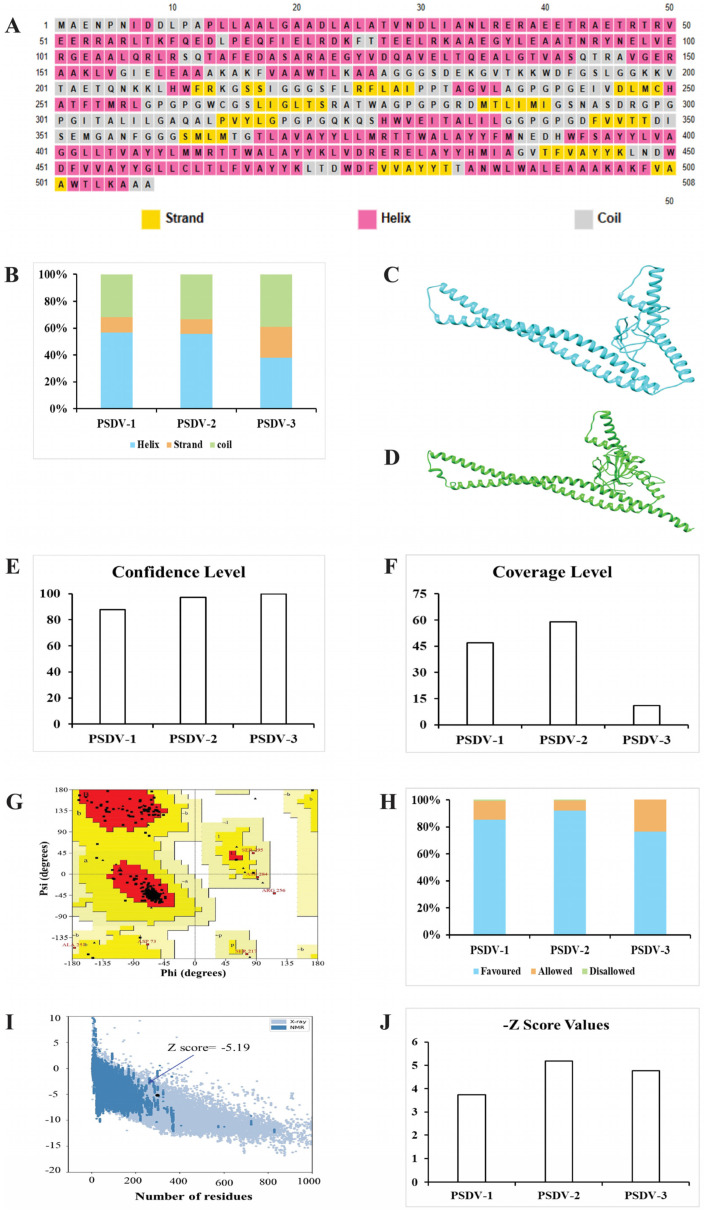
**Modeling and validation of vaccine candidates’ molecular structure.** (**A**) Illustration of the secondary structure for the PSDV-2 vaccine candidate. Strand, helix, and coil are indicated with distinct colors: yellow, pink, and gray, respectively. (**B**) Composition of secondary structure for PSDV-1, -2, and -3, showing the percentage distribution of coils, helices, and β-strands in each protein. (**C**) Predicted tertiary structure: structure of PSDV-2. (**D**) Tertiary structure of the template model (c6tpiA) was used for modeling the PSDV-2 tertiary structure. (**E**) Confidence level for the tertiary structure models of PSDV-1, PSDV-2, and PSDV-3. (**F**) Coverage level of tertiary structure models PSDV-1~3. (**G**) Ramachandran plot of PSDV-2, which emphasizes the residues in favorable A, B, L regions displayed with red colors. (**H**) Distribution of residues in favored, allowed, and disallowed regions of PSDV-1, PSDV-2, and PSDV-3, according to Ramachandran plots. (**I**) Z-score plot of PSDV-2, which was scored as −5.19. (**J**) Z-score of PSDV-1~3, according to their Z-score plots, among which PSDV-2 showed the lowest value.

**Figure 4 biology-13-00681-f004:**
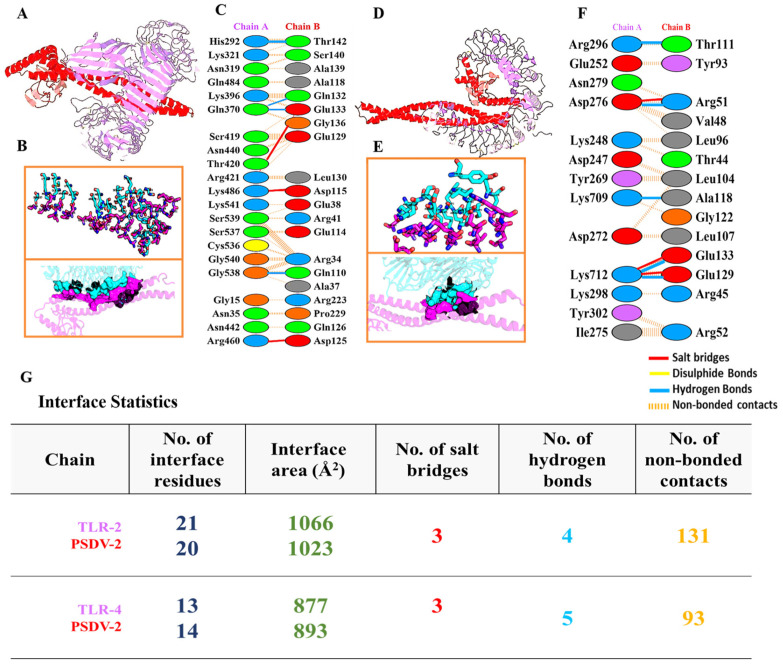
**Molecular docking and interaction analysis of PSDV-2**–**TLR2/4 complexes.** (**A**,**D**) Visualization of the docking between PSDV-2 and TLR2 (**A**) or TLR4 (**D**). PSDV-2 is indicated in red, while TLR2/4 are in indigo. (**B**,**E**) Structure of interaction interface between PSDV-2 and TLR2 (**B**) or TLR4 (**E**). Residues in PSDV-2 are shown in cyan, and residues in TLR2/4 are shown in indigo, which was visualized with Pymol. (**C**,**F**) Detailed schematic of the salt bridges (red), hydrogen bonds (blue), disulfide bonds (yellow), and non-bonded contacts (brown dots) formed in the interface between PSDV-2 with TLR2 (**C**) or TLR4 (**F**). (**G**) Comprehensive statistics for the interface properties between PSDV-2 with TLR2 or TLR4.

**Figure 5 biology-13-00681-f005:**
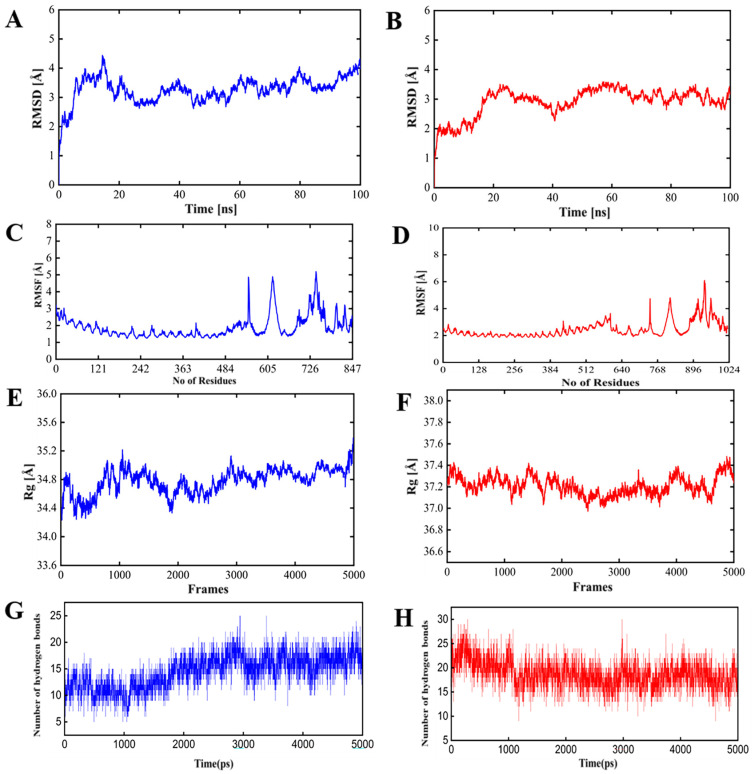
**Molecular dynamics simulation of the PSDV-2**–**TLR2/4 complexes.** (**A**,**B**) Root mean square deviation (RMSD) of the complexes formed with PSDV-2 and TLR2 (**A**) or TLR4 (**B**). (**C**,**D**) Root mean square fluctuation (RMSF) of the complexes formed with PSDV-2 and TLR2 (**C**) or TLR4 (**D**). (**E**,**F**) Radius of gyration (Rg) of the complexes formed with PSDV-2 and TLR2 (**E**) or TLR4 (**F**). (**G**,**H**) Number of intra-molecule hydrogen bonds within the complexes of PSDV-2–TLR2 (**G**) or PSDV-2–TLR4 (**H**). All plots associated with TLR2 are colored in blue, while the ones associated with TLR4 are in red.

**Figure 6 biology-13-00681-f006:**
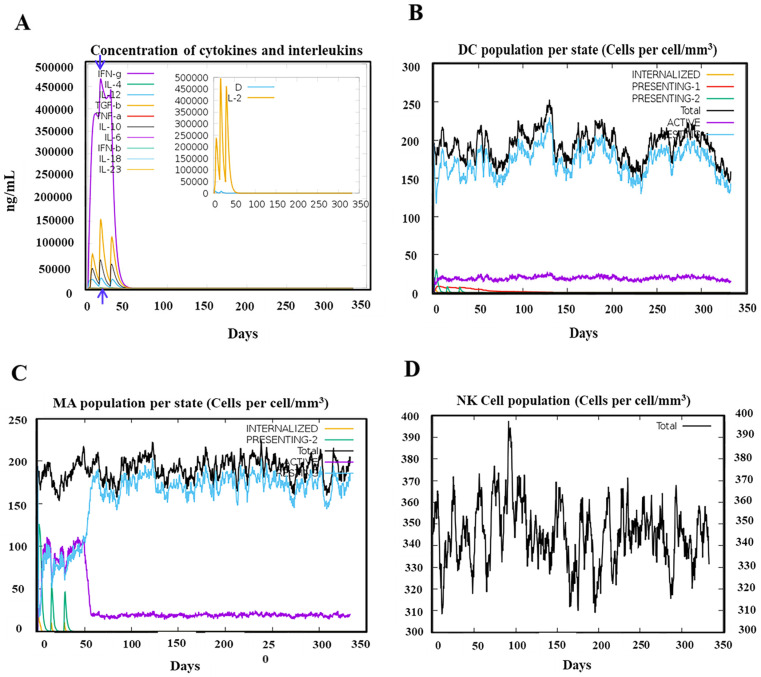
**Simulation of innate immune responses post PSDV-2 vaccination.** (**A**) Kinetics of key cytokine expression levels following three doses of PSDV-2. IFN-gamma, IFN-beta, IL-2, IL-4, IL-6, IL-10, IL-12, IL-18, IL-23, TGF-beta, TNF-alpha were included. Peak level of IFN-gamma was indicated with blue arrow. (**B**) Responses of dendritic cells (DCs) after three doses of PSDV-2. Total population and subpopulations of DCs are presented. The subpopulations included internalized, presenting-1, presenting-2, and active and resting DCs. (**C**) Responses of macrophages (MA) post-vaccination of PSDV-2 with three doses. Total population and subpopulations of macrophages (MA) are illustrated. The subpopulations included internalized, presenting-1, presenting-2, and active and resting MA. (**D**) Post-vaccination dynamics of natural killer (NK) cells.

**Figure 7 biology-13-00681-f007:**
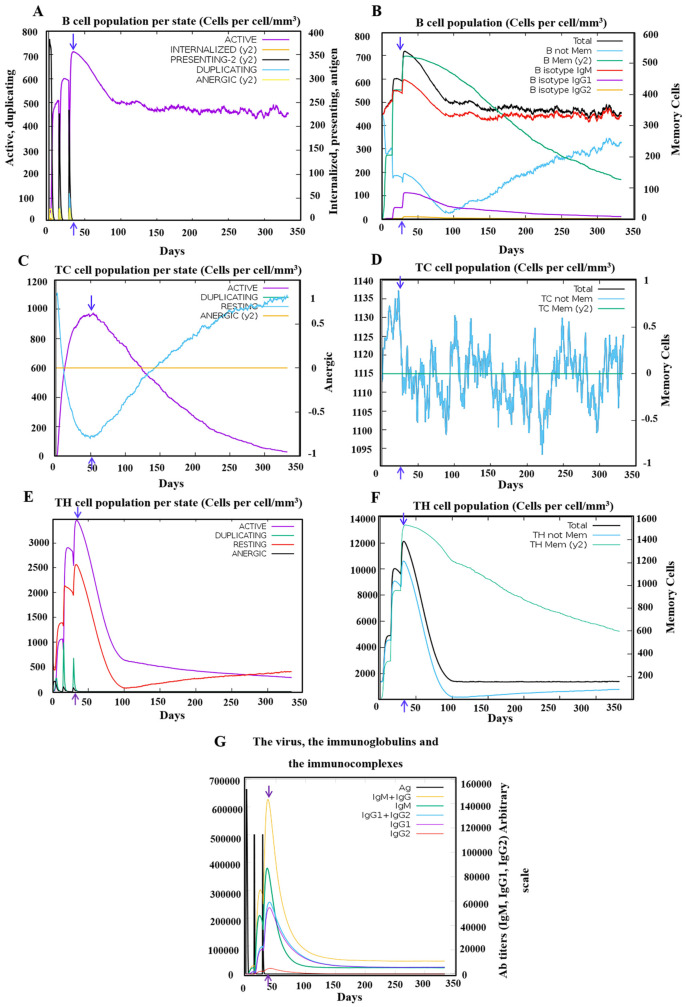
**Simulation of adaptive immune response post PSDV-2 vaccination.** (**A**) States of the B cell population post vaccination. Subpopulations of B cells are presented as active, internalized (y2), presenting-2 (y2), duplicating, and anergic (y2). (**B**) Display of B cell subpopulation based on their status of antibody production. The total population is also presented and separated into B not mem, B mem (y2), B isotype IgM, B isotype Ig1, and B isotype Ig2. (**C**,**D**) Dynamics of CD8+ cytotoxic T cells post-vaccination in distinct activation status (**C**), like active, duplicating, resting, and anergic (y2); and in different memory conditions (**D**). (**E**,**F**) Post-immunization trends of CD4+ helper T cells in distinct status of activation (**E**), including active, duplicating, resting, and anergic (y2), while in different conditions of memory (**F**). (**G**) Production of immunoglobulins along with the administration of three doses of PSDV-2. Antigen (Ag) and a panel of immunoglobulins like IgM + IgG, IgM, IgG1 + IgG2, IgG1, and IgG2 are exhibited. The blue arrows in each plot indicate the peak of immune cells and immunoglobulins.

**Figure 8 biology-13-00681-f008:**
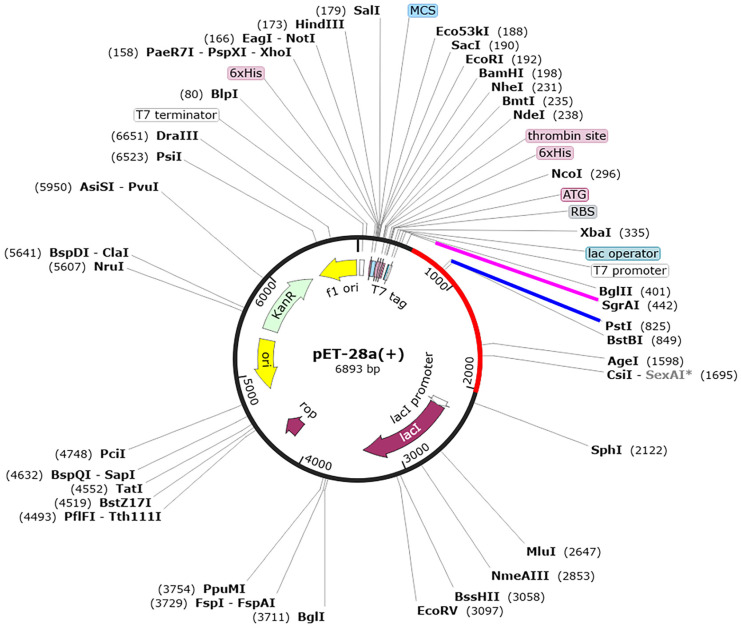
The in silico cloning of plasmid for the recombinant expression of the PSDV-2 vaccine candidate based on the pET28a (+) vector. Blue and pink colour represents “restriction enzyme sites” and Asterisk indicate” physical impact of DNA methylation on enzyme activity.

**Table 1 biology-13-00681-t001:** **Final selection of B-cell epitopes for the vaccine design.** For each selected epitope, the sequence, length, location in the envelope protein, and immunological properties are indicated in the table.

Features	DENV-1	DENV-2	DENV-3	DENV-4
Epitope Sequence	DEKGVT	WDFGSLGG	VTAETQN	LHWFRKGSSI
Start	341	377	330	254
End	346	384	336	263
Protein region	E (ED III)	E (ED III)	E (ED III)	E (ED II)
Epitope length	6	8	7	10
Allergenicity	Non-Allergen	Non-Allergen	Non-Allergen	Non-Allergen
Antigenicity Score	0.7715	2.1175	1.2240	0.9330
Toxicity	Non-Toxic	Non-Toxic	Non-Toxic	Non-Toxic
Intra conservancy	100%	100%	100%	100%
Inter-DENV-1–4 conservancy	25.00% (1/4)	25.00% (1/4)	25.00% (1/4)	25.00% (1/4)

**Table 2 biology-13-00681-t002:** **Final selection of MHC-I CTL epitopes for vaccine design.** For each selected epitope, their corresponding serotype, source protein region, sequence, affinity, immunological properties, and conservancy are presented. The affinity unit is nM.

Serotype	Protein Region	Epitope Sequence	Affinity (nM)	Antigenicity/Immunogenicity Score	Allergenicity/Toxicity Score	Intra-/Cross-Serotype Conservancy
DENV-1	NS1	MLMTGTLAV	3	0.5611/0.20739	Non-Allergic/Non-Toxic	100%/25%
	NS3	LLMRTTWAL	4.24	0.9556/0.27922	Non-Allergic/Non-Toxic	100%/25%
	NS5	FMNEDHWFS	38.43	0.4990/0.40604	Non-Allergic/Non-Toxic	100%/25%
DENV-2	NS1	LVAGGLLTV	39.51	0.5463/0.08268	Non-Allergic/Non-Toxic	100%/25%
	NS3	LMMRTTWAL	4.1	1.1235/0.27922	Non-Allergic/Non-Toxic	100%/25%
	NS5	KLVDREREL	3.1	1.4751/0.09999	Non-Allergic/Non-Toxic	100%/50.00% (2/4),DENV-3,-2
DENV-3	NS1	HMIAGVTFV	5.19	0.8396/0.25559	Non-Allergic/Non-Toxic	100%/25%
	NS3	KLNDWDFVV	4.15	2.2249/0.37972	Non-Allergic/Non-Toxic	100%,25%
DENV-4	NS1	GLLCLTLFV	5.23	0.8159/0.02693	Non-Allergic/Non-Toxic	100%/25%
	NS3	KLTDWDFVV	4.15	2.6071/0.3944	Non-Allergic/Non-Toxic	100%/25%
	NS5	TTANWLWAL	39.89	0.9911/0.42125	Non-Allergic/Non-Toxic	100%/25%

**Table 3 biology-13-00681-t003:** **Final selection of MHC-II HTL epitopes for vaccine design.** For each selected epitope, their corresponding serotype, sequence, MHC allele, source protein region, affinity, immunological properties, and conservancy were presented. The affinity unit is nM.

Serotype	Epitope Sequence	Allele	Protein Region	Affinity (nM)/ Antigenicity Score	Allergenicity/Toxicity	Serotype-Specific Cross Conservancy
DENV-1	FLRFLAIPPTAGVLA	DRB1_0101	Capsid	11.2/0.6827	Non-Allergic/Non-Toxic	100%/50.00% (2/4) DENV-1,-2
DENV-1	EIVDLMCHATFTMRL	DRB1_0701	Capsid	6.9/0.9597	Non-Allergic/Non-Toxic	100%/75.00% (3/4) DENV-1,-2,-3
DENV-2	WCGSLIGLTSRATWA	DRB1_0101	Capsid	7.4/0.8137	Non-Allergic/Non-Toxic	100%/50.00% (2/4) DENV-2,-3
DENV-3	RDMTLIMIGSNASDR	DRB1_0401	Capsid	5.9/0.9291	Non-Allergic/Non-Toxic	100%/25% (1/4)
DENV-4	ITALILGAQALPVYL	DRB1_0101	Capsid	4.7/0.6489	Non-Allergic/Non-Toxic	100%/25% (1/4)
DENV-4	QKQSHWVEITALILG	DRB1_0701	Capsid	12.9/1.2944	Non-Allergic/Non-Toxic	100%/25% (1/4)
DENV-4	DFVVTTDISEMGANF	DRB1_0401	Capsid	33.1/0.6533	Non-Allergic/Non-Toxic	100%/75.00% (3/4) DENV-2,-3,-4

**Table 4 biology-13-00681-t004:** Properties of PSDV-1~3 vaccines in physiocochemistry and immunology.

Parameter	PSDV-1	PSDV-2	PSDV-3
No. of amino acids	564	508	394
Molecular weight	62.09 kDa	55.2733 kDa	42.71375 kDa
Instability Index	26.02	24.28	20.52
Aliphatic index	86.28	88.33	85.05
Half-life	30 h (mammalian reticulocytes, in vitro). >20 h (yeast, in vivo). >10 h (*Escherichia coli*, in vivo).	30 h (mammalian reticulocytes, in vitro). >20 h (yeast, in vivo). >10 h (*Escherichia coli*, in vivo).	30 h (mammalian reticulocytes, in vitro). >20 h (yeast, in vivo). >10 h (*Escherichia coli*, in vivo).
Solubility	0.788949	0.761350	0.569153
Hydropathicity (GRAVY)	−0.049	0.018	0.151
Theoretical pI	8.38	5.76	9.37
Antigenicity	0.6717	0.6727	0.7320
Allergenicity	Non-Allergenic	Non-Allergenic	Non-Allergenic
Toxicity	Non-Toxic	Non-Toxic	Non-Toxic

**Table 5 biology-13-00681-t005:** **Quantitative characterization of interaction between PSDV-2 and TLR4 or TLR2.** Res (Residue), Rec (Receptor, TLR4 or TLR2), Lig (Ligand, PSDV-2), BFE (Binding free energy, unit: kcal/mol), CBFE (Complex binding free energy, unit: kcal/mol).

Immune Receptors	Docking Score	Res (Rec)	BFE	Res (Lig)	BFE	CBFE
TLR4	−5386.23	A-ARG-421	−5.86	B-ARG41	−5.85	−28.13
		A-TYR-297	−3.68	B-ARG45	−4.49	
		A-LYS-462	−2.77	B-GLU-114	−4.15	
		A-TYR-350	−2.5	B-LEU-104	−3.49	
		A-LEU-345	−2.01	B-LEU-107	−3.1	
TLR2	−7330.87	A-ARG-421	−6.24	B-GLU-133	−2.15	−29.66
		A-ARG-460	−4.76	B-GLU-38	−2.14	
		A-GLY-538	−3.68	B-THR-142	−1.96	
		A-SER-539	−3.42	B-GLN-132	−1.86	
		A-HIE-292	−2.78	B-ASP-115	−1.85	

## Data Availability

All data are available in the manuscript and Appendix A.

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
