# Peer review of "An In Silico Design of a Vaccine against All Serotypes of the Dengue Virus Based on Virtual Screening of B-Cell and T-Cell Epitopes"

_biology, 2024, doi:10.3390/biology13090681_

Round 1

Reviewer 1 Report

Comments and Suggestions for Authors

Major concerns:

·       Figures do not have legible font size. Some figures themselves are too small and it’s hard to follow the figures or labels on a printed page.

·       Table headings and Figure legends need to be more descriptive and thoroughly described. For example, legend of Fig1 and Fig2A are not fully descriptive or distinct. Is Fig3A a secondary structure? Of what? Do some of the Tables need a description of the units/scale/score of parameters in footnotes? Lines 346-351, 403, 436: not adequately described.

·       Writing needs to be checked and corrected in many places throughout the text, including in legends as described above, that seems to have a range of issues. For example, line 35: do you mean ‘vaccine candidates’? line 87: “innocent” or naïve? Line 221-223: “studies focused on” seems inadequate to describe in methods what was done. Line 276. Line 445-446: please revise, “fatal symptoms”, “living vaccine”

·       Line 263-264: please describe how the ADE-associated epitopes were determined for exclusion. Mentioning that literature is listed is not sufficient, describe what was done and how.

·       References are missing in many places. For example, lines 29,62,64, 73, 108, 411,460, 465

Comments:

·       Line 465: what did these publications describe?

·       Line 204:  After generation of the residues, how were the rotamers determined and assessed before analyzing interface areas?

·       Line142: please check whether all programs are listed in the supplementary table.

·       Line 281: how was the coverage calculated?

·       Typos or errors: lines 285, 274, 279 (Table S17?). Table at line 285: The last entry in the first column is DENV4 or DENV2?

·       Line 306-311: A brief sentence or two stating which score is desired might be useful.

·       Line 341-342: how is the “best qualified” tertiary structure determined to be the best, based on what?

Comments on the Quality of English Language

 Writing needs to be checked and corrected in many places throughout the text, and the choice of words should be appropriate.

Author Response

Major concerns:

      Figures do not have legible font size. Some figures themselves are too small and it’s hard to follow the figures or labels on a printed page.

Author Response: Thank you for your feedback regarding the figures in the manuscript. To address the  issue with font size and figure clarity, we have enlarged figures where necessary and increased the font size to improve the readability. The updated figures now have been provided in the revised manuscript as well as in Supplementary File 2 for Supplementary Figures.

  • Table headings and Figure legends need to be more descriptive and thoroughly described. For example, the legends of Fig1 and Fig2A are not fully descriptive or distinct. Is Fig3A a secondary structure? Of what? Do some of the Tables need a description of the units/scale/score of parameters in the footnotes? Lines 346-351, 403, 436: not adequately described.

Author Response: Thank you for your valuable feedback regarding the table headings and figure legends. To address these issues, we revised all figure legends and table headings in manuscript and Supplementary File 1 and 2. Adequate number of descriptions had been added to make figure legends and table headings more informative. The units/scale/score of parameters have been added in the tables or described in footnotes. The changes are now listed as below with their current placements in the revised manuscript and supplementary files:

Line 48, for graphical abstract;

Line 131, for the legend of Figure 1;

Line 385, for the legend of Figure 2;

Lines 442, for the legend of Figure 3; Fig3A showed the secondary structure of PSDV-2 vaccine candidate.

Lines 469, for the legend of Figure 4;

Lines 503, for the legend of Figure 5;

Lines 522, for the legend of Figure 6;

Lines 549, for the legend of Figure 7;

Lines 576, for the legend of Figure 8;

Supplementary File 2 contains the legends of Figures S1 to S10;

Line 345, for the heading of Table 1;

Line 362, for the heading of Table 2;

Line 366, for the heading of Table 3;

Line 420, for the heading of Table 4;

Line 478, for the heading of Table 5;

Supplementary File 1 contains the headings of Tables S1 to S23.

  • Writing needs to be checked and corrected in many places throughout the text, including in legends as described above, that seems to have a range of issues. For example, line 35: do you mean ‘vaccine candidates’? line 87: “innocent” or naïve? Line 221-223: “studies focused on” seems inadequate to describe in methods what was done. Line 276. Line 445-446: please revise, “fatal symptoms”, “living vaccine”

Author Response: Thank you for your detailed feedback on the manuscript. We appreciated your observations and made the necessary revisions to improve the clarity and accuracy of the text. We have thoroughly checked and corrected the writing throughout the manuscript, including figure legends and other sections, as needed. The changes are now listed as below with their current placements in the revised manuscript and Supplementary Files 1 and 2:

Line 35 corrected into revised manuscript Line 40, we changed to “three candidate vaccines, PSDV-1, PSDV-2, and PSDV-3”

Line 87 corrected into revised manuscript Lines 88-89, we corrected the “innocent” to “who had no previous dengue infection history.”;

Lines 221-223 corrected into revised manuscript Line 276 we changed “studies focused on” to “ analyze statistical parameters” with which more detail of the method Section 2.5 was provided;

Lines 290-291, Line 276, 445 and 446, these sentences had been modified in better manner way in Lines 371,372 section 3.3, Lines 353,354 section 3.2, Lines 586,587 section 4.

Your feedback is valuable in enhancing the quality of our manuscript, and we have addressed these issues comprehensively in the revised version.

  • Line 263-264: please describe how the ADE-associated epitopes were determined for exclusion. Mentioning that literature is listed is not sufficient; describe what was done and how.

Author Response: Thank you for highlighting the need for a more detailed description regarding the exclusion of ADE-associated epitopes.

A comprehensive explanation had been added in Line 242-246 under the “Materials and Methods” as following: “Exclusion of ADE-Related Epitopes: To mitigate the risk of ADE, relevant literatures were collected and utilized to extract the potentially problematic epitopes, which is detailed in Supplementary Table 11. ADE-related epitopes were then excluded accordingly from the list of predicted B cell epitopes, mainly based on their sequence and location in Envelop protein.”

Moreover, further description was added in Line 628-632 under the “Results” as following: “Further refinement involved the exclusion of epitopes associated with ADE-associated epitopes, as identified in previous studies [55, 56]. The literature supporting this exclusion is listed in Table S11. Attention was paid on the regions within Domains I and II of the Envelope protein [Sarker A, Dhama N, Gupta RD. Dengue virus neutralizing antibody: a review of targets, cross-reactivity, and antibody-dependent enhancement.], which was reported highly correlated with ADE effects. Therefore, the candidate B cell epitopes located in these two regions were largely removed from the epitope list.”

The details can also be seen in Supplementary Table 11 in the Supplementary File 1. The step-by-step procedure of B cell epitope screening was presented in Supplementary Figure 2 in the Supplementary File 2.

References are missing in many places. For example, lines 29, 62, 64, 73, 108, 411,460, 465

Author Response: we have updated the revised manuscript and added the missing references in lines. 64, 65, 75,106, 540, 604,619

Comments:

Comment 1:   Line 465: what did these publications describe?

Author Response: Thank you so much for your valuable question on our manuscript. To answer the question about reference in Line 465, we have included more references of earlier studies in Line 655 of the revised manuscript. In the previous studies about in-silico designing of Dengue vaccines, the researcher conducted their work based on only one selected strain or representative strain of dengue virus, or even targeting a certain structural or non-structural protein encoded by this virus. These strategy restrained the coverage and protectiveness of vaccine candidates developed by them.

On contrary, in our designing, a significantly different strategy was applied. At first, we targeted 10 countries like China, Pakistan, USA, India, Bangladesh, Brazil, Philippines, Singapore, Thailand, and Vietnam as the sources of Dengue polyprotein sequences, based on the high prevalence of dengue virus strains DENV1-4 existing and the data availability in NCBI. Based on large number of full-length Polyprotein sequences retrieved worldwide, conserved fragments were then generated for each serotype and applied in following steps of design. We believe this strategy would be helpful to achieve the new vaccine candidate to provide broad-spectrum protections against numerous reported strains of Dengue Virus globally.

Comment 2:       Line 204:  After the generation of the residues, how were the rotamers determined and assessed before analyzing interface areas?

Author Response: Thank you for your comment regarding the methodology described in line 204. To clarify the process, the PDBSum web server was used for determining the rotamers and automatically generated the side chain confirmations like salt bridges, hydrogen bonds, disulfide bonds. And “interface area of the protein-protein interaction” here means the interaction interface in the complexes of PSDV2-TLRs (TLR2 and TLR4).  After that, we did the Molecular dynamic simulation through AMBER 18 Software to achieve the stability information of these complexes. These descriptions can be seen in Lines 461-467, Figure 4, 482-497, and Figure 5 under section 3.4.

Comment 3:         Line 142: please check whether all programs are listed in the supplementary table.

Author Response: Thank you for your comment in detail of programs. We have carefully reviewed the supplementary table S1 in Supplementary File 1, and we are sure that all programs used in our study have been correctly listed, which included every software tools, databases, and computational resources mentioned in the manuscript. Appropriate details were also incorporated in Supplementary Table S1.

Comment 4:   Line 281: how was the coverage calculated?

Author Response: Thank you for your question regarding coverage calculation. To clarify, coverage was calculated using the Conservancy analysis tool of immune epitope database (IEDB). Detailed description had been added in Line 270-274 of the revised manuscript to provide enough information on how coverage was calculated.

Comment 5:   Typos or errors: lines 285, 274, 279 (Table S17?). Table at line 285: The last entry in the first column is DENV4 or DENV2?

Author Response: Thank you for your points on our typos and errors in writing. We apologize for these mistakes sincerely. The typos and errors had been corrected in the revised manuscript and listed as below:

Line 363, 364, “Tables S13-17 for MHC-I, S18-23 for MHC-II” means Supplementary Tables 13-17 for MHC-I and Supplementary Tables 18-23 for MHC-II. All these Supplementary Tables were integrated in the Supplementary File 1;

Line 379, for the Table 3, extra space in table heading had been removed;

Line 379, the last entry in the first column is DENV4. We had corrected this mistake in revised manuscript.

Comment 6:      Line 306-311: A brief sentence or two stating which score is desired might be helpful.

Author Response: Thanks to the reviewer for highlighting this. Both “Instability Index” and “Aliphatic Index” are important in evaluation of the stability.

Proteins with an Instability Index below 40 are considered stable, so these three PSDV1-3 are all stables. This is advantageous for vaccine formulations as stable proteins are less likely to degrade and lose efficacy over time. Also, a higher aliphatic index is generally desirable and useful. It indicates a higher content of aliphatic side chains, which contributes to the protein's thermal stability, which is important for vaccines to remain effective under varying storage and transport conditions. In our study, this score was even higher than 85 for all three candidates. Especially, PSDV-2 showed the Aliphatic Index 88.33, the highest value among the all candidates, which suggested it as a valuable molecule for further validation and development.

Corresponding sentences had been added, please see Lines 406-415.

Comment 7:    Line 341-342: how is the “best qualified” tertiary structure determined to be the best, based on what?

Author Response: Thank you for your question. The "best qualified" tertiary structure was determined using the Phyre2 tool, which predicts protein structures based on homology modeling. We selected the PSDV-2 structure was selected based on highest coverage score 59%, highest percentage of residues 97.6% in favored (92.0%),highest Z-score value -5.19 as compared PSDV 1 and 2, indicating the best stability and quality. The details are provided in Line 429-440 section 3.4 of the revised manuscript.

Reviewer 2 Report

Comments and Suggestions for Authors

Major issues:

1. The authors were supposed to send the write-up for proofreading and language editing prior to submission. I could not comprehend the writing completely due to its poor language. Worst, it affects my judgement towards the content, therefore, I am not able to provide very objective comments for improvement as I might have mistaken the meaning of the content.

2. Below are a few sincere thoughts about the content based on what I could comprehend from the writing:

i. In your writing you mentioned dengue vaccines are facing infection risks. How could a vaccine face infection risks? Do you mean challenges?

ii. What did you remove from the epitopes in regards to their toxicity and allergenicity? Due to the poor language, I could not understand this part of the writing.

iii. What is the main function of the vaccine? antiviral? immune-boosting?

iv. If the vaccine encourages inflammatory responses, will it exacerbate ADE?

v. If the vaccine only binds to TLRs (innate immunity), how does it trigger memory immunity to protect people from infection s and re-infections?

Comments on the Quality of English Language

Very poor language usage. It requires a major overhaul and professional language editing.

Author Response

Major issues:

Comment 1:   The authors were supposed to send the write-up for proofreading and language editing prior to submission. I could not comprehend the writing completely due to its poor language. Worst, it affects my judgement towards the content, therefore, I am not able to provide very objective comments for improvement as I might have mistaken the meaning of the content.

Author Response: we are thankful to the reviewer for investing the valuable time that you have given to our manuscript. We apologize for any confusion caused by the initial write-up due to language clarity issues. We have carefully revised the manuscript to improve clarity and readability, addressing the concerns you raised. We believe these revisions enhance the overall quality and comprehensibility of the text. The updated manuscript now has been resubmitted for your review. We appreciate your patience and look forward to your feedback on the revised version.

Comment 2:   Below are a few sincere thoughts about the content based on what I could comprehend from the writing:

  1. In your writing you mentioned dengue vaccines are facing infection risks. How could a vaccine face infection risks? Do you mean challenges?

Author Response: Thank you for your comment. You are correct; the term "infection risks" might be misleading. What we intended to convey is that certain vaccines, including those for dengue, face specific challenges, such as reduced effectiveness and the potential for antibody-dependent enhancement (ADE), which can inadvertently increase susceptibility to infections under certain condition. To clarify, vaccines are designed to stimulate an immune response and provide protection against disease. However, in some cases, factors like suboptimal vaccine efficacy or ADE can lead to unintended consequences that may complicate disease outcomes. In this context, "challenges" would be a more accurate term than "infection risks" in Line 19, and Antibody-dependent enhancement (ADE) in Line 31.

In Lines 19,20,30,31 the corresponding changes had been made on our revised manuscript.

Once again, we appreciate your attention to this detail and have made the necessary adjustments to clarify this in the revised manuscript.

  1. What did you remove from the epitopes in regards to their toxicity and allergenicity? Due to the poor language, I could not understand this part of the writing.

Author Response: Thank you for your comment. To clarify, during the vaccine design process, various bioinformatics tools were utilized to address concerns about epitope safety. Specifically, two web servers, Allertop v.2.0 and Toxinpred, were employed respectively to assess the risks of allergenicity and toxicity for the candidate epitopes. After their evaluation, the candidate epitopes will be define as “allergic or non-allergic” and “toxic or non-toxic”. The epitopes defined either as allergic or toxic were then excluded from the list of candidate epitopes, only the ones as both non-allergic and non-toxic were reserved and applied to the following steps of design. This action is necessary and commonly conducted to ensure the safety of vaccine candidates generated in our work and other studies.

In Lines 158,162 section 2.2.1, 187,190 section 2.2.2, 331,332,354,355,356 section 3.2 the corresponding changes had been made on our revised manuscript.

iii. What is the main function of the vaccine? antiviral? immune-boosting?

Author Response: Thank you for your question. The primary function of the multi-epitope-based vaccine, PSDV-2, is to elicit a robust and targetive immune response against Dengue Virus. This type of vaccine is designed to enhance the ability of host immune system to recognize and respond to the pathogen, thereby providing comprehensive protection to the host.

The main functions of the PSDV-2 vaccine include:

  • Immune Boosting: The vaccine was designed to enhance the overall immune response by activating multiple components of the host immune system. This includes the production of antibodies, the activation of T and B cells, and the development of memory T and B cells, all which contribute to a more effective and sustained immune defense against the pathogen.
  • Antiviral Activity: This function is not played by the vaccine directly. But, once the host immune system boosted, antibodies and T cells will work together to block the viral infection and mediate the elimination of virus or infected cells objectively, through which the antiviral activity will be practiced.

In summary, the multi-epitope vaccine PSDV-2 is designed to provide the anti-Dengue Virus protection through immune boosting. This aim will be achieved with the various epitopes derived from the Dengue proteins.

  1. If the vaccine encourages inflammatory responses, will it exacerbate ADE?

Author Response: Thank you for your question. In a nature, inflammatory responses can be part of the downstream effects of ADE, potentially exacerbating the pathogenic condition in secondary infection. To address this problem in our designing, we have excluded epitopes associated with allergies, toxicity, and ADE to minimize the risks of vaccine application.

What’s more, the immune simulation in Fig 6 suggested excessive inflammation can be avoid while strongly boosting the immune response. As showed in Fig 6A, IFN-gamma and IL-2 levels were induced to nearly 500,000, which is much higher than the levels of inflammatory cytokines like IL-6 and TNF-alpha. It’s well known that IFN-gamma plays important role in activation of Macrophage, B and T cells, also works as the activator in anti-viral innate immune; while IL-2 is necessary in anti-infection immune based on its function to promote the maturation and enhancement of T cells.

This result indicated that PSDV-2 in our design can robustly enhance the immune response without promoting excessive inflammatory responses. Our approach is effective to balance the immune activation and safety, which will prevent any exacerbation of ADE ultimately.

  1. If the vaccine only binds to TLRs (innate immunity), how does it trigger memory immunity to protect people from infection s and re-infections?

Author Response: We appreciate your time and concern regarding the study. Toll-like receptors (TLRs) play a crucial role in initiating the innate immune response. The binding to TLRs is necessary in the recognition of foreign pathogens or molecules by the host immune system. Once the vaccine binds to the TLRs, it will be recognized, captured, proceeded and presented by antigen-presenting cells (APCs), primarily like dendritic cells and macrophages. The fragments of vaccine molecule will then be presented to helper T cells, cytotoxic T cells, and B cells as the inducers of the adaptive immune. In this process, memory T and B cells will be generated, which is the natural outcome of immune stimulation.

Also, as indicated in Fig 6, memory T and B cells were generated after 3 doses of immunization and their levels were maintained as long as one year. Once re-infections occur, these immune memory cells will be re-activated for protection.

Since then, the generation of memory immunity can be triggered by our vaccine candidate.

Round 2

Reviewer 2 Report

Comments and Suggestions for Authors

1. The authors improved the language used in the write-up which helped with the review. 

2. The confounding areas are resolved and clarified.